# Impacts of Chemical-Assisted Thermal Pretreatments on Methane Production from Fruit and Vegetable Harvesting Wastes: Process Optimization

**DOI:** 10.3390/molecules25030500

**Published:** 2020-01-23

**Authors:** Ümmihan Günerhan, Ender Us, Lütfiye Dumlu, Vedat Yılmaz, Hélène Carrère, Altınay N. Perendeci

**Affiliations:** 1Istanbul Water and Sewerage Administration, Ataköy Advanced Biological Wastewater Treatment Plant, Şevketiye Street, Havaalanı Avenue, 34156 Istanbul, Turkeyenderus@iski.istanbul (E.U.); 2Ministry of Environment and Urbanization, General Directorate of Environmental Management, Department of Water and Soil Management, 06520 Ankara, Turkey; lutfiye.dumlu@csb.gov.tr; 3Environmental Engineering Department, Artvin Çoruh University, 08100 Artvin, Turkey; vedatyilmaz@artvin.edu.tr; 4Univ Montpellier, INRAE, LBE, 102 Avenue des Etangs, 11100 Narbonne, France; helene.carrere@inra.fr; 5Department of Environmental Engineering, Akdeniz University, 07058 Antalya, Turkey

**Keywords:** biogas production, fruit and vegetable harvesting wastes, process optimization, thermo chemical pretreatment

## Abstract

The increasing population creates excess pressure on the plantation and production of fruits and vegetables across the world. Consumption demand during the whole year has made production compulsory in the covered production system (greenhouse). Production, harvesting, processing, transporting, and distribution chains of fruit and vegetables have resulted in a huge amount of wastes as an alternative source to produce biofuels. In this study, optimization of two pretreatment processes (NaOH and HCl assisted thermal) was investigated to enhance methane production from fruit and vegetable harvesting wastes (FVHW) that originate from greenhouses. NaOH concentration (0–6.5%), HCl concentration (0–5%), reaction temperature (60–100 °C), solid content (1–5%), time of reaction (1–5 h), and mixing speed (0–500 rpm) were chosen in a wide range of levels to optimize the process in a broad design boundary and to evaluate the positive and negative impacts of independent variables along with their ranges. Increasing NaOH and HCl concentrations resulted in higher COD solubilization but decreased the concentration of soluble sugars that can be converted directly into methane. Thus, the increasing concentrations of NaOH and HCl in the pretreatments have resulted in low methane production. The most important independent variables impacting COD and sugar solubilization were found to be chemical concentration (as NaOH and HCl), solid content and reaction temperature for the optimization of pretreatment processes. The high amount of methane productions in the range of 222–365 mL CH_4_ gVS^−1^ was obtained by the simple thermal application without using chemical agents as NaOH or HCl. Maximum enhancement of methane production was 47–68% compared to raw FVHW when 5% solid content, 1-hour reaction time and 60–100 °C reaction temperature were applied in pretreatments.

## 1. Introduction

The acceptance of the generated wastes as a resource within the scope of the circular economy, recovery, and reuse is the major condition for sustaining life on earth. The increasing population coupled with economic development has resulted in huge agricultural production [1]. Based on the consumption requirements, the plantation and production of fruits and vegetables have increased across the world and production reached 1960 million tons in 2017 [2]. The continuation of consumption demand during the whole year has made production compulsory in the covered production system (greenhouse).

China is placed in the first rank of the world with a 27000 km^2^ covered area for the production of fruits and vegetables. South Korea, Spain, and Japan followed China in the ranking. Turkey, holding approximately 772 km^2^ of covered production area, occupies fifth place across the world for the massive production of fresh fruit and vegetables from greenhouses. It is expected that the global commercial greenhouse market will be 32.31 billion USD with an annual growth rate of 8.8% by 2021. Judging by this enormous production, it is predictable that huge amounts of fruits and vegetables wastes have been generated during the production, harvesting, processing, transporting and distribution chain. Fruits and vegetable wastes are the main components of municipal solid waste. The conventional disposal methods of FVHW from greenhouses are the storage in forest area, uncontrolled burning and landfilling by farmers [3]. Moreover, processing, transporting and distribution chain wastes are disposed of landfills by the municipalities. Although EU greenhouse gas emissions from landfills have already declined due to the well-practiced MSW managements [4], the degradation of FVHW during landfilling causes the production of leachates and emissions of greenhouse gases all around the world. Furthermore, uncontrolled burning has also resulted in greenhouse gases and toxic compounds [5].

Maintaining high agricultural production capacities for Turkey, as well as for other countries, has three main objectives. These are increasing agricultural income and its contribution to national income, meeting agricultural and food demand of nationals domestically, and keeping farmers on their lands to secure agricultural production [6]. About 250 km^2^ of greenhouses have been located in Antalya, south of Turkey. It is around 47% of Turkey’s greenhouse production hold in Antalya. Eighty percent of fruit and vegetables, such as tomato, cucumber, pepper, eggplant, courgette, banana, strawberry and any other are grown in the greenhouses. Apart from all these positive contributions, the region has been under heavy pressure in terms of FVHW to be managed. The production area urgently required the implementation of relevant policies for managing the waste disposal and recovering energy from renewable and cost-free lignocellulosic wastes.

On the other side, fossil fuels have supplied over 80% of global energy needs. However, these fuels are not renewable and lead to climate change by the production of CO_2_ and other greenhouse gases [7]. Consequently, building a sustainable society requires reducing dependence on fossil fuels, minimizing the amount of waste produced, and increasing recycling. Generating renewable bioenergy from wastes makes it possible to realize these targets at the same time. The anaerobic digestion (AD) of FVHW combines these advantages, as it produces methane and simultaneously treats the wastes. AD is a trustful and approved technology with a long exercise for the degradation of organic wastes from industry, municipality, and agriculture over the decades. It is known that the predicted energy gain/input ratio for AD was 28.8 MJ/MJ. This finding has been better compared to other technologies for energy production from biomass in energy efficiency [8,9,10]. 

However, the complex structure of agricultural residues is the major limitation, which makes lignocellulosic biomass refractory to anaerobic digestion and yields in low methane production [11,12]. This restricts the hydrolysis of material in which enzymes fail to degrade complex structures. Commercial limitations of methane production from lignocellulosic residues should be solved by the pretreatment processes [12,13]. To solve this limitation, different kinds of pretreatment methods such as mechanical, chemical (acidic, alkali, oxidative), thermal, biological, and combinations have been applied to the countless number of lignocellulosic feedstock [10].

On the other hand, FVHW, primarily non-commercial parts of plants such as leaves, bodies, stems as well as unmerchantable fruits and vegetables, have a milder lignocellulosic structure than the other agricultural residues and require less severe pretreatment conditions before AD. There have been very limited studies investigating the methane production potentials of mixed FVHW combined with pretreatment methods in the literature [3,5,14,15,16,17].

In this study, it was proposed to combine the cost-effective integrated process of chemical-assisted thermal pretreatment and anaerobic digestion for the improvement of methane production from FVHW. The impacts of sodium hydroxide (NaOH) and hydrochloric acid (HCl) assisted thermal pretreatment on methane production were investigated. In the first stage of the work, impacts of variables (reaction temperature, time of reaction, amounts of chemical, solid content, mixing speed) on soluble chemical oxygen demand (sCOD) and soluble sugars (sSugar) were determined with very detailed experimental work for NaOH- and HCl-assisted thermal pretreatments. In the second part of the work, process optimizations were performed for both pretreatment processes; optimum conditions for methane production were discussed and biochemical methane potential (BMP) tests were implemented under the determined conditions. Finally, molecular bond characterization and surface properties of raw and pretreated FVHW were compared. 

## 2. Materials and Methods

### 2.1. Sources of FVHW and Characterization Analysis

FVHW were obtained from Antalya region. The area of Kumluca has been well known for the extensive production of fruits and vegetables in greenhouses. Kumluca area makes a significant contribution to the fruits and vegetables supplies of Turkey and its trade partners during the four seasons. FVHW included leaves; stalks; stems; roots; and unmerchantable fruits from pepper, cucumber, tomato, courgette, and eggplant. FVHW were sun-dried and milled to 1 mm size for the characterization tests, while size-reduced (4–5 mm) fresh wastes were subjected to chemical-assisted thermal pretreatment experiments and BMP tests. The analyses of total solids (TS), volatile solids (VS) and total chemical oxygen demand (COD) were performed according to standard methods 2540C and 5220B, respectively [18]. The contents of lignin, cellulose, hemicellulose, and soluble matter were determined using procedures proposed by Van Soest (1963) [19] by Gerhard FBS6 (Gerhard, Königswinter, Germany). Carbohydrate and the soluble reducing sugar (sRedSugar) concentrations were measured via the Anthrone method [20] and the Dinitrosalicylic acid (DNS) method [21], respectively. Protein concentration was done by the Lowry method [22]. Extractive matter and lipids of the wastes were measured by soxhlet extraction using petroleum ether [23]. Elemental composition of the wastes was determined by CHNS analyzer (LECO, CHNS-932, St. Joseph, MI, USA). The Kjeldahl nitrogen content of the FVHW was quantified by an analyzer (Büchi Digest Automat K-438, Büchi Auto Kjeldahl Unit K-370 and Radiometer TitraLab 840, Büchi, Flawil, Switzerland). 

### 2.2. Chemical-Assisted Thermal Pretreatments Experiments

Fresh FVHW were pretreated in a 0.3 L stainless steel reactor (Parr series 5500 high-pressure reactor; Parr Instrument Company, Moline, IL, USA) equipped with a temperature controller and magnetically-driven turbine agitator. NaOH and HCl were selected as chemical agents in the chemical-assisted thermal pretreatments experiments to enhance methane yield. Selected independent variables and their ranges for NaOH- and HCl-assisted thermal pretreatment are presented in Table 1. For the specified conditions, chemically-assisted thermal pretreatment runs were performed in parallel. NaOH and HCl solutions were loaded into the reactor along with the computed amount of FVHW and heated to predefined reaction temperatures. When the predefined reaction temperature was sustained, the time of reaction was initiated. Later, completing the target reaction time, the reaction vessel was soaked into water and an ice bath to cool down to 40 °C in a short time. Evaluation of the pretreatment performance was made by sCOD, sRedSugar, and BMP. Samples were separated into a liquid and solid phase by centrifugation after cooling. sCOD and sRedSugar analyses were performed from the liquid phase. The analysis of sCOD was performed by a Hach-Lange DR5000 spectrophotometer (Hach Lange GmbH, Duesseldorf, Germany) and a Lange LT200 (Keison Products, Grasshut, Germany) with COD kits. Since the studies done in literature to determine the effectiveness of chemically-assisted pretreatment processes are generally based on the usage of the DNS-reduced sugar method instead of the Anthrone method for the sSugar analysis [24,25,26,27,28], sRedSugar concentrations were quantified by the DNS method [21]. Solubilization degrees for COD and Sugar (or variation in sCOD and sSugar compared to raw FVHW) were calculated according to Mottet et al. (2009) [29]. Pretreated FVHW containing liquid and solid fractions were preserved at −20 °C for the BMP test.

Chemical-assisted pretreatments experiments were planned by central composite design (CCD) of Response Surface Methodology (RSM). Four independent variables with the three levels were utilized for CCD in Design-Expert® 11 software trial version (Minneapolis, MN, USA) for NaOH- and HCl-assisted thermal pretreatments. Fifty-one pretreatment experiments containing three experiments at the center point along with the duplicates of each run were planned for each chemical (NaOH and HCl) assisted thermal pretreatment by CCD. Consequently, 108 values for each response (sCOD, sRedSugar) were used in the optimization of each chemical-assisted pretreatment processes.

Results of experiments were modeled and acquired regression models were assessed by the analysis of variance (ANOVA), regression coefficients along with the *p-* and *F-* values. The quality of models was concluded via the coefficient of determination (*R*^2^) and an adjusted determination coefficient (adj-*R*^2^). With the help of the obtained model equations, optimizations of chemical-assisted thermal pretreatment processes were performed by using maximization and minimization criteria for independent and dependent variables [3,30]. The goal settings of the variables have been adjusted by using the plus (+) symbols in the module of optimization of a Design-Expert^®^ software program for process optimization. The significance of the goals was chosen as (+++++), the highest, for all variables in software. Design-Expert^®^ software solved all goals for all variables to obtain the desirability function. The optimization module investigated the merging of independent variable values that meet the demands placed on each of the dependent variables. The targets were chosen as maximum sSugar and sCOD production, and optimizations were performed for economical process conditions along with the high sSugar and sCOD production.

### 2.3. Biochemical Methane Potential (BMP)

The impact of chemical-assisted thermal pretreatment was also evaluated by a BMP test. The BMP test according to Carrère et al. (2009) and Us and Perendeci (2012) [3,31] was accomplished. Raw and pretreated FVHW, micro and macro nutrients, anaerobic seed sludge along with the buffer solution were incubated at mesophilic temperature (35 °C). Seed sludge was obtained from a wastewater treatment plant’s anaerobic digester unit in Antalya, Turkey. The substrate-to-seed ratio in every reactor was 0.5 (gVS/gVS for samples). After adding the sample, seed sludge, necessary nutrients, and buffer solution to the reactors, volume was set to 400 mL by adding deionized water. pH was set to neutral, and reactors were flushed with N_2_/CO_2_ (70%/30%) gas mixture to provide an anaerobic environment. Triplicate BMP tests were done for all pretreated samples, raw waste, and seed sludge control. BMP tests ended after the 30 days. During fermentation, CH_4_ and CO_2_ components formed in the BMP reactors were detected by Varian CP-4900 Micro-Gas Chromatography (Micro-GC) (Agilent Tech., Santa Clara, USA). Varian CP-4900 was supplied with a thermal conductivity detector (online-TCD) and a 10 m PPQ column. Injector and column temperatures in Micro-GC were 110 °C and 70 °C, respectively. High purity grade helium gas was used as a mobile phase with a flow rate of 25 mL/min. For measuring total gas volume, a gas-liquid displacement device was used filled with a saturated pH 1 acidic salt solution prepared according to Standart Method 2720 [18]. The amount of methane produced from seed sludge used for BMP tests was calculated as mLCH_4_/gVS and then this value was subtracted from the methane amount produced in reactors to normalize the produced methane amount. 

### 2.4. Surface Properties and Bond Structure 

Scanning electron microscopy (SEM) was utilized to evaluate microstructural changes on the surface of wastes after the application of chemical-assisted thermal pretreatments. Raw and pretreated FVHW were lyophilized and coated with gold-palladium for 120 seconds under 18 mA vacuum and samples were examined by SEM with a Zeiss Leo 1430 electron microscope (ZEISS International, Oberkochen, Germany) at a voltage of 15 kV. 

To investigate the impacts of thermochemical pretreatment on waste structure molecular-bond characterization, samples were examined by an FTIR-Varian 1000 Scimitar series FTIR spectrometer (Agilent Tech., Santa Clara, USA). Samples were lyophilized and pelletized with KBr before analyses. Measurements were performed in 500 cm^−1^ and 4000 cm^−1^ wavelength range with a spectral resolution of 4 cm^−1^ and with 16 scans per sample.

## 3. Results and Discussion

### 3.1. Impacts of Chemical Assisted Thermal Pretreatments 

High carbohydrate content as cellulose and hemicelluloses was obtained as 56.86% from the FVHW (Table 2). Carbon content (C%) of FVHW from the elemental analysis was found to be 34.16% (49.1% VS basis). The analysis results suggested that FVHW is a suitable waste substrate for methane production. 

The main purpose of NaOH- and HCl-assisted thermal pretreatment of FVHW is to make more cellulose and hemicellulose available for the anaerobic microorganism and to enhance methane production. Since the enhancement of methane production was related to sCOD and sSugar [3,32], impacts of NaOH- and HCl-assisted thermal pretreatment on sCOD and sSugar were evaluated for the FVHW. The results of the experiments provided a clear indication that NaOH- and HCl-assisted thermal pretreatment boosted a remarkable increase in sCOD concentration, as illustrated in Figure 1a,c. As shown in Figure 1a, an average sCOD concentration of 564 mgsCOD gVS^−1^ was obtained with no NaOH addition, almost at the same concentration level, which varied in the narrow range of 486–629 mgsCOD gVS^−1^, regardless of the changes of other independent variables. On the other hand, increasing NaOH concentration from 0% to 3.3% and 6.5% boosted the COD solubilization. The average sCOD values of 1112 mgsCOD gVS^−1^ and 1243 mgsCOD gVS^−1^ were measured for 3.3% and 6.5% NaOH concentration, respectively, with a moderate fluctuation between 832–1248 mgsCOD gVS^−1^ and 1034–1356 mgsCOD gVS^−1^. Furthermore, no remarkable changes with other independent variables were found. The most significant increase in sCOD of 1356 mgsCOD gVS^−1^ was observed in the pretreatment experiment performed at 6.5% NaOH concentration, 100 °C reaction temperature, 1 h reaction time, and 500 rpm mixing speed, yielding more than a 240% increment compared to raw sCOD of the FVHW. Conclusively, the addition of NaOH resulted in more dissolved organic material, measured as sCOD. Thus, the concentration of sCOD increased with increasing NaOH concentration.

Figure 1c presents that HCl-assisted thermal treatment also raised the sCOD release, but it produced a lower impact on sCOD than by the application of NaOH-assisted thermal pretreatment. When no HCl was applied in the pretreatment experiments, the level of measured sCOD was in the range of 504–613 mgsCOD gVS^−1^ with the average sCOD of 554 mgsCOD gVS^−1^. A similar trend observed in sCOD by increasing the amount of NaOH was observed by increasing the amount of HCl. However, the measured amounts were at the level of 613–1149 mgsCOD gVS^−1^ (average 800 mgsCOD gVS^−1^) and 590–1232 mgsCOD gVS^−1^ (average 908 mgsCOD gVS^−1^), for 2.5% and 5% HCl, respectively, used in the pretreatments. 

Basically, Figure 1a,c indicated that (i) the increase in sCOD obtained by increasing the NaOH concentration was higher than the increase in sCOD measured by increasing the HCl concentration. (ii) The sCOD increment consistently stayed higher when the NaOH was increased to 3.3–6.5% while it remained at the same level when other independent variables were kept in their range. (iii) The amount of sCOD was also increased when the HCl was elevated to 2.5–5%, but apart from the increase in HCl, the increment of sCOD was affected by the dry matter content, reaction temperature and mixing speed. This observation is particularly essential so that the selected chemical and its concentration should be adjusted to solubilize the lignocellulosic structure, but not to oxidize them to produce further metabolites in the process optimization. Furthermore, apart from the selected chemical and the amount applied, each of the other independent variables should also be evaluated in process optimization.

Teghammar et al. (2010) [33] investigated the steam explosion and nonexplosive hydrothermal pretreatment with NaOH for paper tube residuals to improve biogas production. The effect of the reaction temperature, reaction time and NaOH concentration were statistically evaluated on sCOD. sCOD of the untreated paper tube was measured as 262 mg L^−1^, whereas sCOD of the nonexplosively pretreated paper tube at 190 °C with 2% NaOH was found at 2280 mg L^−1^. The results from Teghammar et al. (2010) [33] revealed that increasing the NaOH concentration yielded high sCOD production, which is somehow coupled with high phenolic compounds solubilization. Wang et al. (2009) [34] applied hydrothermal alkaline pretreatment to sorted municipal solid waste to enhance biogas production. sCOD concentration increased from 5931 mg L^−1^ at 1 g NaOH/100 g solid to 12007 mg L^−1^at 4 g NaOH/100 gr solid. They indicated that the concentration of sCOD increased with the addition of NaOH. Jard et al., 2013 [35] investigated the effect of different thermochemical pretreatments on the solubilization and anaerobic degradability of the red macroalgae *Palmaria palmata*. They also indicated that thermo-chemical pretreatment induced a change in the structure of macroalgae, increasing the cell-wall porosity and thus facilitating the solubilization of COD. The results from our study were found to be consistent with Wang et al. (2009), Teghammar et al. (2010) and Jard et al. (2013) [33,34,35].

The NaOH- and HCl-assisted thermal pretreatment experiments also pursued to observe the alteration of soluble reducing sugars (Figure 1b,d). The soluble reducing sugar profile displayed in Figure 1b clearly explains the impact of NaOH-assisted thermal pretreatment; with the caustic addition, produced sugar from cellulose and hemicellulose was transformed to further metabolites. The highest sRedSugar concentration was measured as 335 mg glucose gVS^−1^ from the FVHW pretreated at 0% NaOH concentration, 100 °C reaction temperature, 4 h reaction time, and 0 rpm mixing speed, where the maximum increase in reducing sugar was calculated as 67.2%. On the other hand, soluble sugar concentration increased by increasing the HCl concentration from 0% to 2.5%, but the sugars produced by increasing HCl dosage to 5% were transformed. Figure 1d apparently shows the impact of HCl-assisted thermal pretreatment: With HCl addition, thermal treatment at a temperature of 100 °C and 3% dry matter content greatly intensified its solubility to 831 mg glucose gVS^−1^ and resulted in a sharp decrease to the average level of 68 mg glucose gVS^−1^ when the HCl concentration was raised to 5%. It should be remembered that the decrease in sugar concentration necessitates the optimization of the amount of chemical usage in the pretreatment process. 

### 3.2. Optimization of Chemical Assisted Thermal Pretreatment Process for sCOD and sSugar

RSM and CCD were applied to derive optimum conditions effective on FVHW for NaOH and HCl-assisted thermal pretreatment. Statistical evaluation was made and the models for sCOD and sSugar were constructed for both pretreatments. Analysis of variance (ANOVA) was employed to investigate the quality of model equations. Quadratic regression models for sCOD and sSugar were significant, as apparent from the Fisher’s *F*-test with the low value (P model > *F* = 0.0001) for NaOH- and HCl-assisted thermal pretreatments. To check the significance of the fit of equations for sCOD and sSugar of NaOH- and HCl-assisted thermal pretreatments, ANOVA was made and results are presented in Table 3 and Table 4, respectively.

The quality of the fit of models was expressed by the coefficient of determination *R*^2^ and adjusted determination of coefficient, *Adj-R*^2^. *R*^2^ and *Adj-R*^2^ were found to be 0.9503 and 0.9325, and 0.9288 and 0.9033 for sCOD, indicating a high degree of correlation between the response and the independent variables in the quadratic models for NaOH- and HCl-assisted thermal pretreatments, respectively. Signal-to-noise ratio is defined as adequate precision and a ratio bigger than four is favorable. Therefore, the ratios 20.06 and 22.33 have shown adequate signals for the models. Values of Prob *> F* less than 0.05 indicated that model terms are significant [36]. While the linear effect of reaction temperature (*A*), mixing speed (*C*) and NaOH concentration (*D*), interactive terms of *AD*, and square terms of reaction temperature (*A*^2^) and NaOH concentration (*D*^2^) were significant for the sCOD model of the NaOH-assisted thermal pretreatment, the linear effect of reaction temperature (*A*), HCl concentration (*B*) and solid content (*C*), interactive terms of *AB* and *BC,* and square terms of reaction temperature (*A*^2^) and mixing speed (*D*^2^), were significant for the sCOD model of HCl-assisted thermal pretreatment. 

Similarly, the values of *R*^2^ and *Adj-R*^2^ were computed as 0.9967 and 0.9956, and 0.9161 and 0.8860 for quadratic models of sSugar of NaOH- and HCl-assisted thermal pretreatments, respectively. Adequate precision with the values of 69.18 and 16.58 was obtained, indicating models can be applied in the design space. Furthermore, the linear effect of reaction temperature (*A*), reaction time (*B*), and NaOH concentration (*D*); interactive terms of *AD* and *BD;* and square terms of NaOH concentration (*D*^2^) were significant for the sSugar model of NaOH-assisted thermal pretreatment, while the linear effect of reaction temperature (*A*), HCl concentration (*B*) and solid content (*C*); interactive terms of *BC;* and square terms of HCl concentration (*B*^2^) were significant for the sSugar model of HCl-assisted thermal pretreatment. 

On the other hand, other variables shown in ANOVA analysis presented in Table 3 and Table 4 were not significant (*p* > 0.005). 

sCOD and sSugar models were operated for the optimization of NaOH- and HCl-assisted thermal pretreatment conditions by Design-Expert® software. During the optimizations of both pretreatment applications, a cost-driven approach was applied. Reaction temperature, reaction time and NaOH concentration were diminished, while sCOD and sSugar were maximized for NaOH-assisted thermal pretreatment. Mixing speed was kept in the ranges. Optimum conditions were obtained with a desirability of 0.681 at 0.67% NaOH concentration, 65 °C reaction temperature, 1 h reaction time, and 500 rpm mixing speed. Under these conditions, 688 mgsCOD mgVS^−1^ and 222 mgsSugar mgVS^−1^ were predicted for sCOD and sSugar, respectively. To validate the optimization, pretreatment experiments were executed under these proposed conditions. In these runs, 712 mgsCOD mgVS^−1^ and 236 mgsSugar mgVS^−1^ were realized for sCOD and sSugar models, respectively. Solubilization of sCOD and sSugar compared to the raw FVHW were found to be 80% and 18%, respectively, for cost optimization conditions.

To optimize the HCl-assisted thermal pretreatment, initial solid loading, sCOD and sSugar were maximized, whereas reaction temperature and HCl concentration were minimized from the point of environmentally sustainable procedures and process costs. Since mixing speed was found to be insignificant for both models, it was set within the range. Optimum conditions were obtained with the desirability of 0.647 at 1.44% HCl concentration, 60 °C reaction temperature, 5% solid content, and 324 rpm mixing speed. Under these conditions, sCOD and sSugar values were estimated as 682 mgsCOD mgVS^−1^ and 553 mgsSugar mgVS^−1^, respectively. To validate the optimization, specific batch runs were conducted. sCOD and sSugar were determined as 698 mgsCOD mgVS^−1^ and 438 mgsSugar mgVS^−1^, verifying the acceptable performance of the models for HCl-assisted thermal pretreatment. The increments of sCOD and sSugar compared to the raw wastes were computed as 76% and 119%, respectively for cost optimization conditions.

The impacts of independent variables of NaOH-assisted thermal pretreatment on sCOD and sSugar solubilization are demonstrated in Figure 2a,b. Figure 2a explains the individual impact of each independent variable on sCOD. Approximately, 700 mgsCOD mgVS^−1^ can be produced when 1-hour reaction time, 0.67% NaOH concentration, 500 rpm mixing speed, and 65 °C reaction temperature conditions were applied in the pretreatment. On the other hand, increasing NaOH concentration resulted in higher sCOD solubilization of approximately 1200–1300 mgsCOD mgVS^−1^. However, as it is apparently observed from Figure 2b, increasing the NaOH concentration yielded extremely decreasing sSugar solubilization that can be converted directly into methane. Approximately, 200–250 mgsSugar mgVS^−1^ should be produced under the same pretreatment conditions.

The independent variables’ impact on sCOD and sSugar solubilization after the application of HCl-assisted thermal pretreatment is also presented in Figure 2c,d. The individual impact of each independent variable on sCOD presented in Figure 2c shows that 600–700 mgsCOD mgVS^−1^ should be produced by the application of 5% solid content, 1.44% HCl concentration, 324 rpm mixing speed, and 60 °C reaction temperature conditions in the pretreatment. As it is clearly seen from Figure 2d, with the increasing HCl concentration, sSugar concentration was reduced to further compounds by the transformation and obtained very low sugar concentration in the pretreatment medium.

The most important difference between the two pretreatments was the use of different initial solid content in the pretreatment. Enhancement of methane production from FVHW can be achieved by the usage of 5% solids content along with the 60–80 °C reaction temperature and no or very low chemical agent in the pretreatment.

### 3.3. Methane Production

Since increasing the NaOH and HCl concentrations from 0% to 6.6% and 0% to 5%, respectively, yielded with the lowly-soluble sugar concentration due to degradation of the produced sugar and possibly the conversion to refractory compounds or inhibitors, the methane productions were investigated with a reduced number of pretreatment experiments. A BMP value of raw FVHW without pretreatment, serving as a comparison, was found as 217 mLCH_4_ gVS^−1^. The observed methane production profiles for thermal and NaOH- and HCl-assisted thermal pretreatments are presented in Figure 3a,b.

Figure 3a,b shows that thermal pretreatments with no addition of chemicals led to the highest methane productions. Methane production was attained in the range of 222–365 mLCH_4_ gVS^−1^ (av. 295 mLCH_4_ gVS^−1^) at a range of 60–100 °C reaction temperature and 1–5% solid content. The highest BMP values were 365 and 360 mLCH_4_ gVS^−1^ acquired at 60 °C and 100 °C reaction temperature, respectively, along with 5% solid content and 1 h reaction time, as indicated in the red dash line zone.

The lowest BMP value (121 mLCH_4_ gVS^−1^) was measured from the FVHW pretreated at a NaOH concentration of 6.5%, a reaction temperature of 100 °C, and a reaction time of 4 h. By adding NaOH as 3.3% and 6.5% concentrations, the production of methane decreased from 251–360 mLCH_4_ gVS^−1^ (av. 305 mLCH_4_ gVS^−1^) without NaOH to 132–133 mLCH_4_ gVS^−1^ (av. 133 mLCH_4_ gVS^−1^) at 3.3% and further dropped to 121–129 mLCH_4_ gVS^−1^ (av. 127 mLCH_4_ gVS^−1^) at 6.5%. As it is easily captured from Figure 3a, pretreatment experiments without NaOH showed an average of 41% enhancement in methane production; on the other hand, an average 40% decrease was observed in methane production with the increased NaOH concentration. The increased concentration of NaOH resulted in low methane production due to degradation of the produced sugar and possible conversion to refractory compounds or inhibitors [37,38,39] or increased concentration of Na^+^ itself or combined effects of both. In fact, for the sake of clarity, Na^+^ concentration was approximately 1.8 g Na^+^/L in the BMP reactor, it reached 2.5 g Na+/L with the supplied NaHCO_3_ as a buffer in the BMP reactor for the lowest BMP obtained as 121 mLCH_4_ gVS^−1^. Despite the fact that slight inhibition concentrations for Na^+^ are in the range of 3.5–5.5 g/L [40], it may be concluded that two effects (degradation of sugar to refractory compounds or inhibitors and Na^+^ concentration) caused inhibition.

As for HCl-assisted thermal pretreatment, minimum methane of 166 mLCH_4_ gVS^−1^ was produced from the wastes pretreated at 2.5% HCl concentration, 60 °C reaction temperature, and 3% solid content. As it is clearly seen from Figure 3b, methane production decreased when the HCl concentration was increased. However, the decrease in methane production was not at the same level as the decrease in NaOH application. Methane production decreased to an average value of 172 mLCH_4_ gVS^−1^ when the amount of HCl was increased to 2.5% and the number of solids was decreased to 3%. On the contrary to this, methane production increased to an average value of 240 mLCH_4_ gVS^−1^ with increasing solids content to 5% and in the range of 60–100 °C reaction temperature. While an average of 31% enhancement was observed in methane production in the pretreatment experiments where HCl was not used, an average of 21% decrease occurred in methane production when 2.5% HCl was added. 

Conclusively, (i) a higher amount of methane production was obtained by the thermal application without using the chemical agent of NaOH or HCl. (ii) Although the chemical usage contributes to the hydrolysis rate of the lignocellulosic structure, the selected high dose for NaOH (3.3–6.5%) and HCl (2.5–5%) should be the probable cause of low methane production. (iii) Maximum enhancement of methane production was 47–68% compared to raw FVHW when 5% solid content, 1 hour reaction time and 60–100 °C reaction temperature were used in pretreatments.

Song et al. (2014) [41] examined the impacts of seven chemical pretreatments on methane production from corn straw. Four acid reagents (H_2_SO_4_, HCl, H_2_O_2_, and CH_3_COOH) at concentrations of 1, 2, 3, and 4% and three alkali reagents (NaOH, Ca(OH)_2_ and NH_3_.H_2_O) at concentrations of 4, 6, 8, and 10% were applied in the pretreatments at 25 °C during the 7 days. Results revealed that acid and alkaline treatments increased methane production by approximately 10.3% to 115.4% higher yield compared to raw. This phenomenon can be explained by the fact that alkaline and acid pretreatments promote organic solubilization and increase the surface area available for enzymatic accessibility. Monlau et al., (2012) [42] applied five thermo-chemical pretreatments by using NaOH, H_2_O_2_, Ca(OH)_2_, HCl, FeCl_3_ at 4–10 gr/100gr TS concentration along with the 30–170 °C reaction time and 1–24 h reaction time to sunflowers stalks. They found that methane potentials of pretreated sunflower with different chemicals increased to 3–35% compared to the raw sunflower. Although corn straw and sunflower stalks do not seem to be similar to the FVHW at first glance, the cellulose and hemicellulose contents of the sunflower are close to FVHW used in this study to make a comparison. Furthermore, the enhancement of methane production results from this study was found to be coherent with the literature and revealed that pretreatment conditions necessitate optimization and depend on the structure and composition of lignocellulosic wastes. As a result, the selection of a wide of range levels of the independent variables has riveting characteristics to optimize the process in a broad design boundary and to evaluate the positive and negative impacts of independent variables along with their ranges. The results obtained from optimization in the design space have resulted in the suggestion of a low amount of NaOH and HCl usage due to the inhibition effect of the high amount of chemicals on methane production. The selection of a wide range of levels of independent variables has provided an opportunity for evaluating extreme points in optimization.

### 3.4. Surface Modification and Molecular Bond Changes in FVHW by Chemical Assisted Thermal Pretreatment

SEM images and FTIR spectra of FVHW pretreated at optimized conditions for sCOD and sSugar by NaOH-assisted thermal pretreatments (0.67% NaOH, 65 °C, 1 h, 500 rpm) and HCl-assisted thermal pretreatments (1.44% HCl, 60 °C, 5% DM, 324 rpm), and maximum CH_4_ production conditions (100 °C, 0% NaOH, 1 h, 0 rpm and 60 °C, 0% HCl, 5% DM, 0 rpm) compared to those of the raw FVHW are demonstrated in Figure 4a,b, respectively.

SEM images presented in Figure 4a,b depicted that both NaOH- and HCl-assisted thermal pretreatments caused modification on the surfaces of FVHW. The images of raw wastes revealed that lignocellulosic material had a smooth and fluent fibrous structure on its surface before any pretreatment. Despite the robust structure of lignocellulosic material, both NaOH- and HCl-assisted thermal pretreatments hydrolyzed lignocellulosic material and destructed the smooth outer surface by opening it up and damaged the inner structure of tissues. SEM images remarked consistent results with methane production from FVHW.

In the dashed line zone of the FTIR spectra, both NaOH- and HCl-assisted thermal pretreated wastes presented a different spectrum than from raw lignocellulosic wastes. Especially, modifications in spectra at 400–500, 780, 875, 1000–1100, 1150, 1245, 1310, 1380, 1500–1600, 1750, 2300–2400 and 400–500, 775, 875, 1000–1100, 1150, 1280–1310, 1380, 1500–1600, 1750, and 2300–2400 cm^−1^ wavelengths can be apparently observed from the pretreated samples of FVHW by the application of NaOH- and HCl-assisted thermal pretreatments, respectively. Most of the observed modifications in the molecular bonds were found to be similar. While FTIR band intensities were closer to each other at 60–100 °C and 0–0.67% NaOH pretreatment conditions and gave lower density than the raw sample, FTIR band intensities of samples pretreated at 1.44% HCl condition were found to be closer to the raw sample.

At 470 and 775 cm^−1,^ FTIR spectra show vibrations of Si–O–Si and NH_2_ bonds, respectively [43,44]. Si may come from small amounts of soil adhering to the roots of FVHW. Absorbance at 875 cm^−1^ wavelength represents the vibration of glucosidic linkage in hemicellulose [45], and it can be concluded that all processed wastes had the same absorbance value, indicating a transformation in the chemical bond structure caused by pretreatment. Bond characteristic changes in samples belonged to C–O stretching vibrations in cellulose/hemicellulose and the aryl-OH group in lignin at 1050 cm^−1^ [46], C–O–C vibrations at β-glucosidic linkages in cellulose and hemicellulose at 1150 cm^−1^ [47], and C–O adsorption at 1245 cm^−1^ [48]. Observed changes at 1310 cm^−1^ and 1380 cm^−1^ of FTIR spectra indicated CH_2_ wagging in cellulose and hemicellulose and planar C–H bending in cellulose, hemicellulose, and lignin, respectively [45]). Since wavelengths between 1500–1600 cm^−1^ are attributed to aromatic C=C bonds, changes at 1510 cm^−1^ were found to belong to aromatic ring vibrations of lignin [49]. As changes at 1734 cm^−1^ pointed to C=O stretching vibration in acetyl groups of hemicellulose [47], FTIR spectra observed at 1750 cm^−1^ may belong to this group. Out of the dashed line zone, pretreated wastes showed a difference only at one wavelength (2400 cm^−1^) with low densities, compared to raw FVHW. The band at 2400 cm^−1^ was attributed to C=O stretching within the carbon dioxide [50].

In summary, FTIR analysis revealed that pretreatment caused changes in molecular bond characterization especially in the region of 800–1500 cm^−1^.

## 4. Conclusions

FVHW collected from the extensive production greenhouses area of Antalya city was thermochemically pretreated to enhance methane production. The most significant increase in sCOD of 1356 mgsCOD gVS^−1^ was observed at 6.5% NaOH concentration, 100 °C reaction temperature, 1 h reaction time, and 500 rpm mixing speed, yielding more than a 240% increment compared to raw sCOD of the FVHW. The highest sSugar concentration was measured as 335 mg glucose gVS^−1^ from the FVHW pretreated with no chemicals at a 100 °C reaction temperature, 4 h reaction time, and no mixing, where the maximum increase in reducing sugar was calculated as 67.2% compared to raw FVHW. Increasing NaOH and HCl concentrations resulted in higher sCOD but extremely lower sSugar solubilization. Results of sCOD and sSugar from the NaOH- and HCl-assisted thermal pretreatment experiments were modeled, and statistically significant regression models were developed. The most important independent variables were found to be chemical concentration (as NaOH and HCl), solid content and reaction temperature for the pretreatment processes. 

Two-hundred-and-twenty-two to 365 mLCH_4_ gVS^−1^ methane productions were obtained by the simple thermal application without using NaOH or HCl. Maximum enhancement of methane production was 47–68% compared to raw FVHW at 5% solid content, 1-hour reaction time and 60–100 °C reaction temperature. 

The optimization of chemically-assisted thermal pretreatment processes should be done with the methane production results obtained from the BMP test. On the other hand, increasing the NaOH and HCl concentrations from 0% to 6.6% and 0% to 5%, respectively, resulted in low soluble sugar concentrations. Furthermore, optimization in the design space suggested the use of low amounts of NaOH and HCl due to the inhibition risk of the high amount of chemicals. In addition, BMP tests yielded to low methane production for the samples pretreated with high a amount of chemicals (NaOH and HCl). As a result, it was found that sSugar and methane production results were correlated and sSugar data can be used in optimization in a reliable way for this substrate.

Results from this comprehensive study have been encouraging that it serves information to build the potential application and to reduce the waste-related environmental pressure in the production-dense area. Gained optimization solution in the design space resulted in the suggestion of no chemical usage due to the inhibition effect of the high amount of chemicals on methane production. The selection of a wide range of levels of independent variables provided an opportunity for evaluating extreme points in optimization.

## Figures and Tables

**Figure 1 molecules-25-00500-f001:**
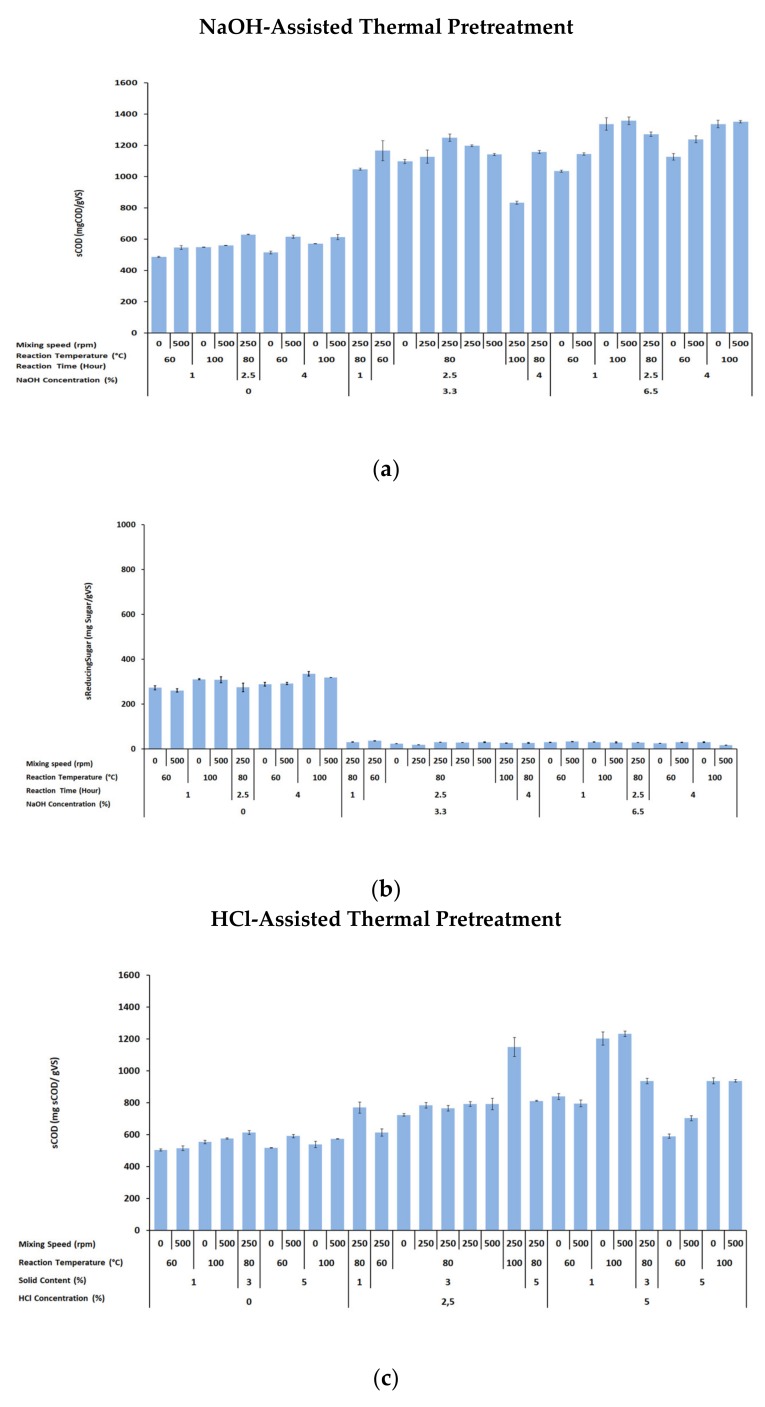
Effects of NaOH-assisted thermal pretreatment on sCOD (**a**) and sRedSugars (**b**); and effects of HCl-assisted thermal pretreatment on sCOD (**c**) and sRedSugars (**d**).

**Figure 2 molecules-25-00500-f002:**
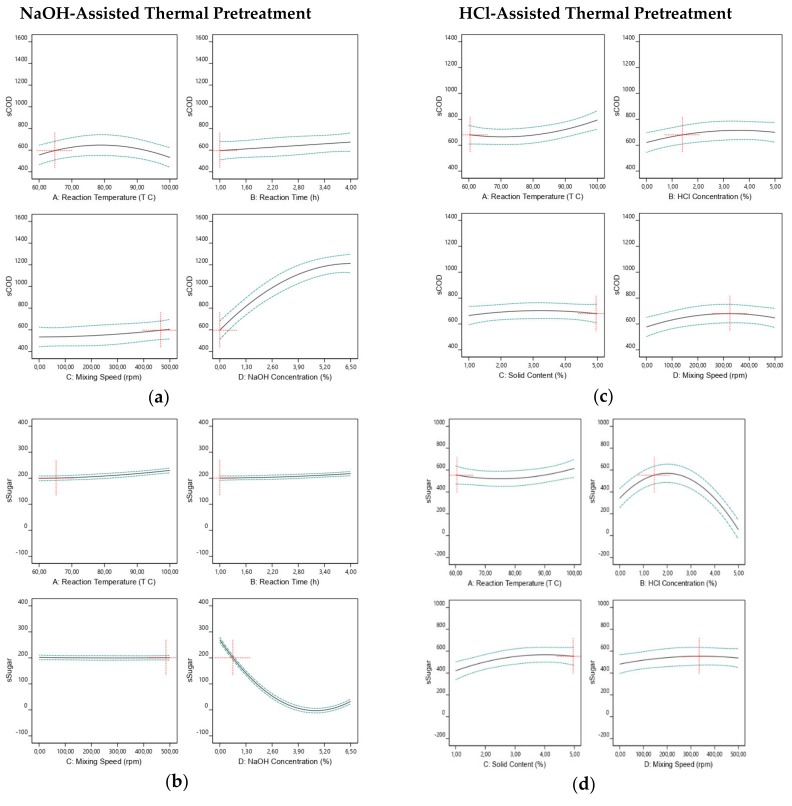
Individual impacts of independent variables (Reaction temperature, reaction time, mixing speed and NaOH concentration) on sCOD (**a**) and sSugar (**b**) of NaOH-assisted thermal pretreatment; individual impacts of independent variables (Reaction temperature, HCl concentration, solid content and mixing speed) on sCOD (**c**) and sSugar (**d**) of HCl-assisted thermal pretreatment.

**Figure 3 molecules-25-00500-f003:**
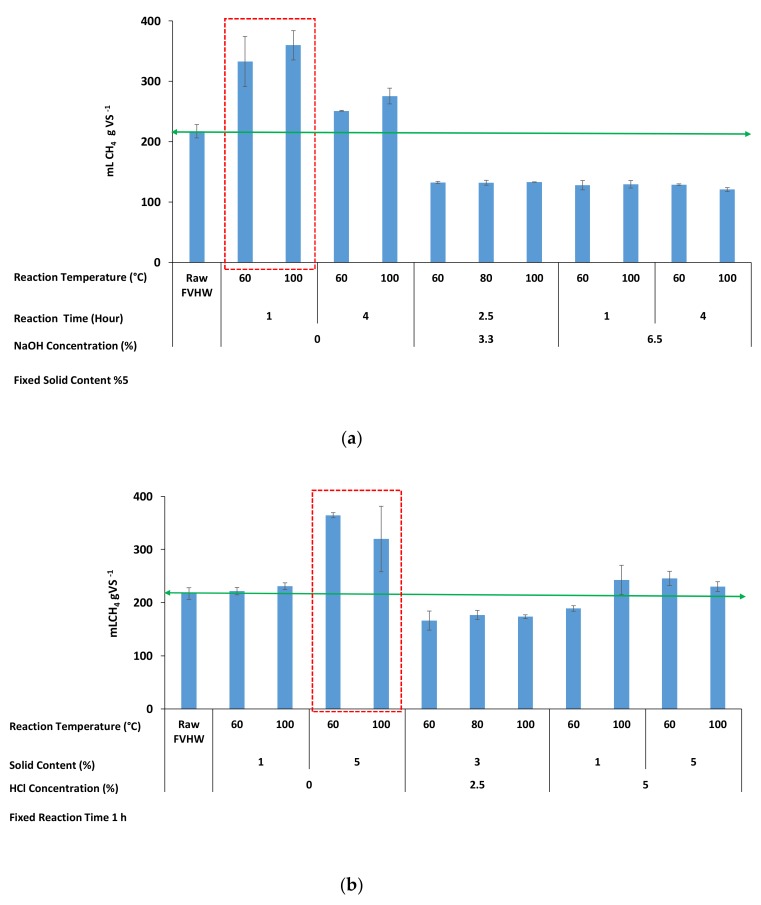
Methane production profiles after the NaOH-assisted thermal pretreatments (**a**); HCl-assisted thermal pretreatments (**b**).

**Figure 4 molecules-25-00500-f004:**
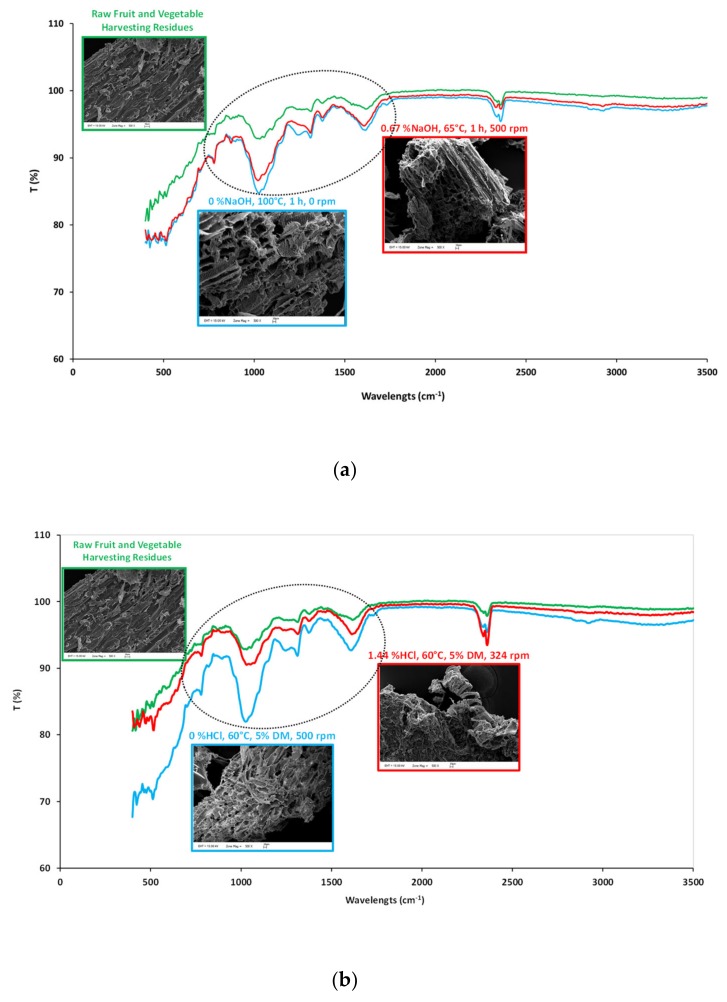
SEM images and FTIR spectra of raw and pretreated FVHW at 0% NaOH concentration, 100 °C reaction temperature, 1 h reaction time and no mixing, and 0.67% NaOH concentration, 65 °C reaction temperature, 1 h reaction time and 500 rpm mixing speed conditions (**a**); SEM images and FTIR spectra of raw and pretreated FVHW at 0% HCl concentration, 60 °C reaction temperature, 5% DM content and 500 rpm mixing speed and 1.44 %HCl concentration, 60 °C reaction temperature, 5% DM content and 324 rpm mixing speed conditions (**b**).

**Table 1 molecules-25-00500-t001:** Selected independent variables and their ranges for NaOH- and HCl-assisted thermal pretreatments.

NaOH-Assisted Thermal Pretreatment			
**Independent Variables**	**Coded Variable Levels**		
	**Minimum (−1)**	**Center (0)**	**Maximum (+1)**
A: Reaction Temperature, (°C)	60	80	100
B: Reaction Time, (Hour)	1	2.5	4
C: Mixing Speed, (rpm)	0	250	500
D: NaOH Concentration, (%, *w/v*)	0	3.25	6.5
E: Dry Matter (DM) Content, (%)	5		
**HCl-Assisted Thermal Pretreatment**			
**Independent Variables**	**Coded Variable Levels**		
	**Minimum (−1)**	**Center (0)**	**Maximum (+1)**
A: HCl Concentration, (%, *w/v*)	0	2.5	5
B: Reaction Temperature, (°C)	60	80	100
C: Dry Matter (DM) Content (%)	1	2.5	5
D: Mixing Speed, (rpm)	0	250	500
E: Reaction Time, (Hour)	1		

**Table 2 molecules-25-00500-t002:** Characterization analysis results of FVHW.

Parameters	Value
Total Solid, TS, g TS/kg dry sample	913.93
Volatile Solid, VS, g VS/kg TS	694.09
Total Kjeldahl Nitrogen, TKN, g N/kg VS	24.80
Total Organic Carbon, g TOC/kg VS	501.77
Soluble Protein (g SProtein/kg VS)	94.56
Soluble Sugar (g SGlucose/kg VS)	279.50
Extractable Material and Lipid (g/kgVS)	42.17
Van Soest Fractionation (TS basis)	
Celulose, %	33.12
Hemicellulose, %	23.74
Lignin, %	9.32
Soluble matter, %	33.82
Elemental Analysis (TS basis)	
Carbon, C, %	34.16
Hydrogen, H, %	5.03
Nitrogen, N, %	2.39
Sulphur, S, %	0.82
C/N	14.29

**Table 3 molecules-25-00500-t003:** ANOVA results for chemical-assisted thermal pretreatments for sCOD.

**NaOH-Assisted Thermal Pretreatment**					
**Source**	**Sum of Square**	**Degrees of Freedom**	**Mean Square**	***F*-Value**	**Prob > *F***
**sCOD**					
Model	4.907 × 10^6^	14	3.505 × 10^5^	53.31	<0.0001
A—Reaction Temperature	43898.63	1	43898.63	6.68	0.0136
B—Reaction Time	23782.27	1	23782.27	3.62	0.0646
C—Mixing Speed	29589.73	1	29589.73	4.50	0.0403
D—NaOH Concentration	4.147 × 10^6^	1	4.147 × 10^6^	630.71	<0.0001
AB	5767.11	1	5767.11	0.8772	0.3547
AC	10885.13	1	10885.13	1.66	0.2058
AD	62377.77	1	62377.77	9.49	0.0038
BC	553.20	1	553.20	0.0841	0.7733
BD	3.86	1	3.86	0.0006	0.9808
CD	245.92	1	245.92	0.0374	0.8476
A²	54067.88	1	54067.88	8.22	0.0066
B²	0.3052	1	0.3052	0.0000	0.9946
C²	1573.40	1	1573.40	0.2393	0.6274
D²	1.185 × 10^5^	1	1.185 × 10^5^	18.02	0.0001
*R*^2^ = 0.9503, *Adj-R*^2^ = 0.9325, *Adeq Precision* = 20.0622 *C.V.* % = 8.33					
**HCl-Assisted Thermal Pretreatment**					
**Source**	**Sum of Square**	**Degrees of Freedom**	**Mean Square**	***F*-Value**	**Prob > *F***
**sCOD**					
Model	2.104 × 10^6^	14	1.503 × 10^5^	36.35	<0.0001
A—Reaction Temperature	4.591 × 10^5^	1	4.591 × 10^5^	111.09	<0.0001
B—HCl Concentration	1.132 × 10^6^	1	1.132 × 10^6^	273.80	<0.0001
C—Solid Content	69230.38	1	69230.38	16.75	0.0002
D—Mixing Speed	10722.26	1	10722.26	2.59	0.1153
AB	2.004 × 10^5^	1	2.004 × 10^5^	48.49	<0.0001
AC	13407.44	1	13407.44	3.24	0.0794
AD	604.22	1	604.22	0.1462	0.7043
BC	1.182 × 10^5^	1	1.182 × 10^5^	28.60	<0.0001
BD	216.27	1	216.27	0.0523	0.8203
CD	5219.10	1	5219.10	1.26	0.2680
A²	18583.84	1	18583.84	4.50	0.0404
B²	11063.97	1	11063.97	2.68	0.1099
C²	4855.75	1	4855.75	1.17	0.2851
D²	20711.11	1	20711.11	5.01	0.0310
*R*^2^ = 0.9288, *Adj-R*^2^ = 0.9033, *Adeq Precision* = 22.3369, *C.V.* % = 8.53					

**Table 4 molecules-25-00500-t004:** ANOVA results for chemical-assisted thermal pretreatments for sSugar.

**NaOH-Assisted Thermal Pretreatment**					
**Source**	**Sum of Square**	**Degrees of Freedom**	**Mean Square**	***F*-Value**	**Prob > *F***
**sSugar**					
Model	8.671 × 10^5^	14	61936.12	854.31	<0.0001
A—Reaction Temperature	2058.59	1	2058.59	28.40	<0.0001
B—Reaction Time	360.68	1	360.68	4.98	0.0315
C—Mixing Speed	83.78	1	83.78	1.16	0.2890
D—NaOH Concentration	6.434 × 10^5^	1	6.434 × 10^5^	8874.32	<0.0001
AB	42.60	1	42.60	0.5876	0.4480
AC	117.12	1	117.12	1.62	0.2112
AD	3549.03	1	3549.03	48.95	<0.0001
BC	8.04	1	8.04	0.1109	0.7409
BD	1264.29	1	1264.29	17.44	0.0002
CD	50.10	1	50.10	0.6911	0.4109
A²	207.77	1	207.77	2.87	0.0985
B²	67.99	1	67.99	0.9378	0.3388
C²	12.02	1	12.02	0.1657	0.6862
D²	82123.05	1	82123.05	1132.76	<0.0001
*R*^2^ = 0.9967, *Adj-R*^2^ = 0.9956, *Adeq Precision* = 69.1844, *C.V.* % = 7.24					
**HCl-Assisted Thermal Pretreatment**					
**Source**	**Sum of Square**	**Degrees of Freedom**	**Mean Square**	***F*-Value**	**Prob > *F***
**sSugar**					
Model	2.318 × 10^6^	14	1.656 × 10^5^	30.43	<0.0001
A—Reaction Temperature	30752.88	1	30752.88	5.65	0.0224
B—HCl Concentration	2.588 × 10^5^	1	2.588 × 10^5^	47.56	<0.0001
C—Solid Content	43307.69	1	43307.69	7.96	0.0075
D—Mixing Speed	8252.21	1	8252.21	1.52	0.2255
AB	2294.35	1	2294.35	0.4217	0.5199
AC	45.89	1	45.89	0.0084	0.9273
AD	512.64	1	512.64	0.0942	0.7605
BC	1.395 × 10^5^	1	1.395 × 10^5^	25.64	<0.0001
BD	1547.35	1	1547.35	0.2844	0.5969
CD	1193.16	1	1193.16	0.2193	0.6422
A²	19378.84	1	19378.84	3.56	0.0666
B²	6.606 × 10^5^	1	6.606 × 10^5^	121.41	<0.0001
C²	21405.59	1	21405.59	3.93	0.0544
D²	7628.98	1	7628.98	1.40	0.2435
*R*^2^ = 0.9161, *Adj-R*^2^ = 0.8860, *Adeq Precision* = 16.5818, *C.V.* % = 26.19					

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
