# Peer review of "Impacts of Chemical-Assisted Thermal Pretreatments on Methane Production from Fruit and Vegetable Harvesting Wastes: Process Optimization"

_molecules, 2020, doi:10.3390/molecules25030500_

Round 1

Reviewer 1 Report

Please do not use the present tense.

“fruit and vegetable harvesting wastes” – I suggest to insert the abbreviation because this statement appears very often in the text.

20 -  “Continuous consumption” – I suggest to change it.

22 – „chain” – plural version.

23 – „be alternative sources to produce biofuels” – stylistics.

30-32 – “Thus, 30 the increased concentration of NaOH and HCl in the pretreatments resulted in low methane 31 production.” - the sentence is incomprehensible.

36-38 “Maximum enhancement of methane production was 47-68% compared to raw fruit and vegetable harvesting wastes when 5% solid content, 1 hour reaction time and 60-100°C reaction temperature is used in pretreatments.” - stylistics, please use the past tense.

39 – “Chemical assisted thermal pretreatment” - in my opinion this is not a proper Keyword.

40 – It is worth adding to Keywords „biogas” and „food waste” in my opinion.

48-49 – “Based on growing requirements, the plantation and production of fruits and vegetables are being 48 enlarged across the world, production reached to 1811 million tons in 2013 (FAOStat, 2015).” - the sentence is incomprehensible.

53 – “have followed” ???

53-55 “Turkey occupies a place in the world for the massive generation of fresh fruit and vegetables from greenhouses; holding approximately thousand decares of covered production area, Turkey takes the fifth place across the world.” - I suggest not using long sentences.

58 – “wastes” – I suggest using waste even in the plural.

66-67 - Please avoid repetitions in the text. In this case it was "production".

77 – „On the other side, fossil fuels have supplied over the 80% of global energy needs.” – in this case, References should be provided.

79-80 – “Consequently, building a sustainable society requires reducing dependence on fossil fuels and reducing the amount of waste produced.” – stylistics.

85 – “MJ/MJ.” Isn't it better to write without a unit if it is a ratio?- This value is given without the unit and can be marked [-].

104 – „methane generation” - if you have "methane production" in Abstract, please write it consistently.

141 „were” not „are”.

Author Response

Please consider the attached file. 

Reviewer 2 Report

I addition to the comments in the enclosed file (please report them in a file in order to give  a point by point answer), please try to highlight the novelty and significance of this paper. It deals with a well known topic

Author Response

Please consider the attached file.

Round 2

Reviewer 1 Report

-

Reviewer 2 Report

Some of the main issues remain and cannot be fixed.